# Memory Savings by Sharing One Source: Insights into Subsetsum Approximation

## Abstract

Large deep neural networks, often fine-tuned from foundation models, dominate modern machine learning, but their high memory requirements limit deployment on resource-constrained devices. Strong lottery tickets (SLTs) offer a promising solution by significantly reducing memory usage, as they are fully characterized by a seed for generating a random source network and a binary mask. Notably, multiple models can share the same source network without increasing its width requirement. As we show, this source sharing can lead to memory savings when experts share specific sparsity patterns. Based on novel insights into optimized subsetsum approximations, we also show how the masks can be adjusted to further reduce memory overhead. To validate these theoretical findings, we provide explicit SLT constructions in experiments.

## 1 Introduction

The rapid advancements in deep learning have led to the widespread development of large neural network models, particularly those fine-tuned from foundation models (Bommasani et al., 2022). Such models, which dominate many areas of modern machine learning, are often tasked with solving highly specialized problems. However, their immense memory requirements pose a significant challenge, especially when deploying these models on resource-constrained devices. This is particularly relevant in real-world scenarios, where numerous models might need to interact and collaborate to solve complex tasks. As we move toward an era where multiple AI models must work together seamlessly, the cumulative memory footprint of these models becomes a critical bottleneck.

While each model might be specialized to a given task, many are derived from the same or similar foundation models, resulting in structural similarities. Such similarities might also occur by design when some of the model components solve tasks jointly like mixtures of experts (MoEs) (Jiang et al., 2024; Lepikhin et al., 2020) or ensembles (Tabik et al., 2020; Kilimci et al., 2018). We argue that structural overlaps can be leveraged to optimize memory usage across multiple models or their components and propose concrete strategies to do so in the context of Strong Lottery Tickets (SLTs).

SLTs can naturally reduce memory footprints even of single models (Ramanujan et al., 2020; Otsuka et al., 2024), as they only need to save a seed for generating a random source network and a binary mask that selects a subnetwork. An overlooked fact is that this principle (Lueker, 1998) encompasses also model families. Multiple models can share the same source network without demanding any increase of the width requirement in state-of-the-art lottery ticket existence constructions (Pensia et al., 2020; Burkholz, 2022b; Ferbach et al., 2022). Sharing a source network has several advantages: a) Multiple models can access the same memory of the source network and potentially reuse computations at (parallel) inference time. b) Saving source model weights instead of a seed could solve issues with reproducibility on different devices. In that case, storing a single source is preferable over multiple ones. c) Beyond the overlap of sparsity patterns of target networks, we can optimize subsetsum approximations to gain more memory-efficient mask representations. Sharing the same source increases such memory efficiency.

Our analysis is based on novel insights into subsetsum approximation, which forms the basis of SLT constructions and defines the masks that approximate given target models. Utilizing the fact that subsetsum approximations have multiple solutions among which we can choose different candidates, we discuss alternative selection criteria. The most memory-efficient approach is neither just maximizing the overlap of the masks nor their individual sparsity. This implies a trade-off between

the objective to reduce memory and inference speed, which would both be desirable in practice. Furthermore, perhaps counter-intuitively, larger source models support more memory-efficient representations, thus, inducing a trade-off between the size of the source model and the masks and the joint memory requirements of the masks.

Overall, our **contributions** can be summarized as follows:

- Up to our knowledge, we are the first to identify the potential of the Strong Lottery Ticket (SLT) concept to reduce the memory footprint of neural network model families like ensembles or mixtures of experts (MoEs).
- We identify multiple advantages of sharing a single source network across multiple models. In particular, we prove that sharing a source network induces a lower width requirement (on the source) in lottery ticket existence statements.
- We obtain conceptual and theoretical insights into subsetsum approximations and utilize them to optimize memory usage across structurally similar models.
- We identify fundamental trade-offs between memory and inference speed in joint SLT constructions of multiple models.

## 2 RELATED WORK

**Ensembles and mixture of experts** The main application of our theory is representing and saving families of models, such as ensembles (Ganaie et al., 2021) and mixture of experts (MoEs) (Cai et al., 2024; Jacobs et al., 1991). Traditional ensemble methods such as bagging (Breiman, 1996), stacking Wolpert (1992) and boosting (Freund & Schapire, 1996) have been applied to deep ensembles (K. et al., 2005; Liu et al., 2014; Low et al., 2020). leading to better generalization performance, efficiency, and less need for stronger models. Nevertheless, neural networks incur high training costs. Proposed alternatives are homogeneous and heterogeneous ensembles, which distribute the data set over the models or use a diverse set of (smaller) models (Tabik et al., 2020; Kilimci et al., 2018). A modern approach is a mixture of experts (MoEs) architecture, contributing to the scaling and success of Large Language Models (LLMs) (Jiang et al., 2024; Lepikhin et al., 2020). However, while MoEs aim to improve inference efficiency, the increasing size of MoE leads to storage issues, for example, when the device capacity is constrained (e.g. in the case of edge devices). Recently, hierarchical storage solutions have been proposed to alleviate this issue (Yu et al., 2024; Hwang et al., 2024; Yi et al., 2023). For example, off-loading inactive experts with selection forecasting has the goal of minimizing the overhead from data transfer. Moreover, specific GPU kernels have been designed to address the issue of memory movement and computational efficiency (Rajbhandari et al., 2022; He et al., 2021; Nie et al., 2022; Hwang et al., 2023). To add to this, PIT (Zheng et al., 2023) introduces a sparse tiling mechanism based on permutation equivariance leading to computational savings.

**Model compression** Besides infrastructural improvements, dense-to-sparse model training (Komatsuzaki et al., 2023), sparse-to-dense model training (Fedus et al., 2022b) and merging experts (Li et al., 2022) also lead to inference and memory savings by reducing the size of the involved models. Model families can also be sparsified jointly according to Allingham et al. (2023). Yet, up to our knowledge, the potential of SLTs to further reduce the memory footprint of such models, has not been explored. Utilizing SLTs, we show it is possible to save multiple models with one seed and their respective binary masks. Furthermore, we show that the storage of the masks can be optimized based on novel insights into subsetsum approximation.

**Strong lottery tickets** Many neural network compression algorithms are inspired by the lottery ticket hypothesis (Frankle & Carbin, 2019), which conjectures the existence of subnetworks of randomly initialized neural networks, the source network, which can approximate a target network if trained in isolation. Ramanujan et al. (2020) found empirical evidence for an even stronger hypothesis, the SLT, that there exist subnetworks that do not require any training but pruning the source network is sufficient to solve complex tasks. Malach et al. (2020) proved this hypothesis formally but required a high overparameterization, namely that the source network is wider than a target network by a polynomial factor in the approximation error, which can yield potentially optimally sparse lottery tickets Fischer & Burkholz (2022). Two subsequent works brought this factor down to logarithmic overparameterization (Orseau et al., 2020; Pensia et al., 2020) suggesting

practical relevance of SLT constructions, which could be exploited for reduced memory representations, as mentioned by Otsuka et al. (2024). (Pensia et al., 2020) have achieved their practical bound by utilizing an older theoretical result on subsetsum approximations (Lueker, 1998) in their SLT construction. Subsequent work has extended this proof strategy to cover modern deep neural network architectures with nonzero biases (Fischer et al., 2021), convolutional layers (da Cunha et al., 2022; Burkholz, 2022b) and residual blocks, (Burkholz, 2022b), various activation functions (Burkholz, 2022c), more general equivariant architectures including graph neural networks (Ferbach et al., 2022), specific universal architectures (Burkholz et al., 2022), and construct structured SLTs or more flexible approximations (Xiong et al., 2023). Also, the distribution of source networks has been generalized to binary (Diffenderfer et al., 2021; Zhou et al., 2019; Sreenivasan et al., 2022) or randomly sparse (Gadhikar et al., 2023) weights. Our results apply to all of these architectures and conditions, as we base our insights on the underlying subsetsum approximation.

**Subsetsum approximation**    Subsetsum approximation is a fairly old result that states that any target variable in a bounded interval can be approximated as the sum of a subset of a sufficiently large source size (Lueker, 1998), which can be used to prove lottery ticket existence (Pensia et al., 2020). This statement has been generalized to larger target intervals (Burkholz et al., 2022), multi-dimensional targets and sources (Becchetti et al., 2022), randomly dropped source elements (Gadhikar et al., 2023), epsilon perturbations (Xiong et al., 2023), and partially frozen sources (Otsuka et al., 2024). In this work, we analyze the properties of the approximating subsets and exploit our flexibility in choosing candidates that reduce the joint memory requirements of multiple targets.

# 3    SLT EXISTENCE WITH SHARED VERSUS SEPARATE SOURCES

Our first objective is to prove the existence of strong lottery tickets (SLTs) for a family of target models such as MoEs and ensembles. A largely overlooked fact is that current lottery ticket constructions already support simultaneously all possible target models that are smaller than a specific upper bound on their size (width and depth) if they share the same random source model, which is pruned to approximate our target models. As a consequence, we only have to store the corresponding masks of each target model and the seed of the random source model generation.

Thus, we only have to compare this setting to the alternative, where we use separate source networks for each expert model. At first glance, separate sources could have the advantage that the corresponding models can be developed independently and do not require synchronization of multiple servers, for instance, if they are trained on different machines (and data potentially). Saving multiple seeds for different source models incurs only negligible additional memory and the computational overhead of generating multiple source models can be parallelized.

So why do we still propose to share the same source network? As we prove, the first disadvantage of separate sources is that this increases our bound on the source width. Thus, each of the multiple source models would need to be larger – to ensure the success of a higher number of subsetsum approximations in the independent LT constructions. However, the width factor is only logarithmic in the number $m$ of experts $\log(m)$, but this can still be significant if our target experts are large. More importantly, we show that sharing a source model allows us to find more memory-efficient lottery tickets. Furthermore, overlapping target sparsity patterns can become more easily exploited. To back these statements up theoretically, we present an analytically tractable comparison of subsetsum representations of multiple targets that highlights the memory benefits of sharing the same source. Moreover, we show that sparsifying each target mask individually is suboptimal and propose another criterion to find more memory efficient representations of multiple targets.

**SLT existence model family without source width increase**    In the most efficient SLT constructions, each target parameter $z$ is approximated by solving an independent subsetsum approximation problem of the following form:

**Theorem 3.1** *(Subsetsum approximation, Theorem 3 Pensia et al. (2020)) Let the source set elements $X_1, X_2, \ldots, X_n$ be i.i.d. uniformly distributed over $[-1, 1]$. Furthermore, set $C = \log e + 1 \approx$*

2.442695. *Suppose that $n \geq C \log \frac{2}{\min\{\epsilon, \delta\}}$. Then, with probability at least $1 - \delta$, we have*

$$\forall z \in [-\frac{1}{2}, \frac{1}{2}], \qquad \exists S \in [n] \text{ s.t. } |z - \sum_{j \in S} X_j| < \epsilon.$$

Originally, this result has been proven by Lueker (1998) and first applied to SLTs by Pensia et al. (2020). An often overlooked fact, which is critical for us, is that the approximation works for **all target** values in $[-\frac{1}{2}, \frac{1}{2}]$. In consequence, Theorem 3.1 applies also to a family of targets. However, if we rely on independent source networks for each target separately (as common in the approximation of the parameters of a single target network), our bound on the source widths increases, as the next theorem establishes.

**Theorem 3.2** *Let $z_1, \ldots, z_m \in [-\frac{1}{2}, \frac{1}{2}]$. The probability of successfully approximating each target $z_i$ based on $m$ independent source sets $(X_1^{(i)}, X_2^{(i)}, \ldots, X_n^{(i)})$ is at least $1 - \delta$ if $n \geq C \log \frac{2m}{\min\{\epsilon, \delta\}}$.*

Proof. Denote the failure probability of single source model of the multiple source case with $\delta' > 0$ and the standard single source case with $\delta > 0$. We want to show that $(1 - \delta')^m \geq 1 - \delta$, for some $\delta' > 0$. We use that for $\delta' \in (0, 1)$ the following inequality holds $(1 - \delta')^m \geq 1 - \delta' m$. This can be shown using induction with $m \geq 1$. The inequality implies that $\delta' \leq \frac{\delta}{m}$. Plugging this into Theorem 3.1 we get

$$n \geq C \log \frac{2}{\min\{\epsilon, \delta'\}} \geq C \log \frac{2m}{\min\{\epsilon, \delta\}},$$

which concludes the proof. $\square$

Theorem 3.2 translates directly to SLT constructions of families of full target networks. Any SLT construction of the modern architectures, which we have discussed in the literature section, can be transferred to our setting with the updated subsetsum approximation. To give an example, we transfer the construction by Pensia et al. (2020) to the following family $\{f_k\}_{k=1}^m \in \mathcal{F}_\ell$ of target multi-layer perceptrons (MLPs) of depth $\ell$, where :

$$\mathcal{F}_\ell := \{f : f(x) = W_\ell \sigma (W_{\ell-1} \ldots \sigma (W_1 x)), \forall i W_i \in \mathbb{R}^{d_i \times d_{i-1}} and ||W_i|| \leq 1\}$$

Note that these weights can also be partially zero or encode identity mappings of specific layers, effectively covering also neural networks of smaller width and depth. The next result finally states how our width requirement increases if we use multiple source networks instead of a single one as basis for $m$ target models.

**Theorem 3.3** *Let $\{f_k\}_{k=1}^m \in \mathcal{F}_\ell$ be a family of target models. Consider $m$ randomly initialized $2\ell$-layer source MLPs $g^k(x) := \tilde{W}_{2\ell}^k \sigma \left( \tilde{W}_{2\ell-1}^k \ldots \sigma \left( \tilde{W}_1^k x \right) \right)$, where every weight is sampled iid from $U[-1, 1]$, $\tilde{W}_{2i}^k$ has dimension $d_i \times d_i'$, and $\tilde{W}_{2i-1}^k$ has dimension $d_i' \times d_{i-1}$. Then, with probability at least $1 - \delta$, for every $f_k \in \mathcal{F}_\ell$*

$$\min_{M_i^k \in \{0,1\}^{|M_i^k|}} \sup_{||x|| \leq 1} ||f_k(x) - M_{2\ell}^k \odot \tilde{W}_{2\ell}^1 \sigma \left( M_{2\ell-1}^k \odot \tilde{W}_{2\ell-1}^1 \ldots \sigma \left( M_1^k \odot \tilde{W}_1^1 x \right) \right) || < \epsilon, \quad (1)$$

*if $d_i' \geq C d_{i-1} \log \frac{d_{i-1} d_i \ell}{\min\{\epsilon, \delta\}}$ for a shared source network. However, the same statement holds for multiple source networks*

$$\min_{M_i^k \in \{0,1\}^{|M_i^k|}} \sup_{||x|| \leq 1} ||f_k(x) - M_{2\ell}^k \odot \tilde{W}_{2\ell}^k \sigma \left( M_{2\ell-1}^k \odot \tilde{W}_{2\ell-1}^k \ldots \sigma \left( M_1^k \odot \tilde{W}_1^k x \right) \right) || < \epsilon, \quad (2)$$

*if $d_i' \geq C d_{i-1} \log \frac{d_{i-1} d_i \ell}{\min\{\epsilon, \delta/\boldsymbol{m}\}}$.*

Proof. The proof is analogous to the on of Theorem 1 (Pensia et al., 2020) but replaces the subsetsum approximation width bound by Theorem 3.2. $\square$

We conclude that reusing the same source network (with index 1) can be achieved by an overparameterization of the form $d' \sim C d \log(1/\delta)$. In contrast, using $m$ independent sources results in $d' \sim C d \log(m/\delta)$. While the logarithmic dependence on $m$ suggests that the approach can be scaled to many experts, the difference $C d \log(m/\delta) - C d \log(1/\delta)$ is still non-negligible if the target dimension $d$ is large. Yet, the following advantages of source sharing are potentially more considerable.

**Saving memory using one source** In addition to decreasing the source width requirement, we can save memory if the masks $M^1, \ldots, M^m$ can be stored efficiently. Next, we show that, if the target networks have similar entries and can be approximated by the same subset, then their masks are identical for such entries if we use the same source. But for different sources, even if one is only a shuffled variant of the other, it is unlikely that the representing masks have such a high overlap. This overlap is relevant for finding masks that can be stored efficiently. We will later propose a scheme where we separately save the overlap of masks and their individual differences from such overlaps. With such an encoding, highly similar targets would have more memory-efficient SLT masks if they are based on the same source network.

**Theorem 3.4** *Let $z_1, z_2 \in [-\frac{1}{2}, \frac{1}{2}]$ be similar targets that can be approximated by $z$ with $|z_i - z| < \epsilon$ and $|z| > \epsilon$ with subsetsum approximation $\sum_{i \in M} X_i = z$ of size $|M| = k$. If the same source is used, we have $M^1 = M^2$. Furthermore, if we reshuffle the sources that are used to approximate individual targets, the probability of perfect overlap between mask entries of $M^1$ and $M^2$ is*

$$\mathbb{P}\left(M^1 \cap M^2\right) = \frac{1}{\binom{n}{k}}. \tag{3}$$

Proof. We are utilizing subsetsum approximations as in Theorem 3.1. The first statement follows from the fact that the targets are identical. Thus, if we use the same source network, both targets have the same subsetsum representation and it is sufficient to save one representation and their (trivial) difference.

The second statement follows from the probability of permutations. We have $k$ options for the first overlap and $n$ total options, next, we have $k - 1$ options and $n - 1$ total options until we have 1 option and $n - k$ total options. This results in the probability being equal to Eq. (3).

Note that if the target $|z| < \epsilon$, we could approximate targets by 0 and thus an empty mask. Irrespective of the source set, such representations would perfectly overlap and have the sparsest possible mask. $\square$

Theorem 3.4 discusses an edge case, when single-source SLTs could be preferable over multi-source SLTs. Instead of complete overlap for a shared source, there is a low probability of overlap in the case of separate sources. Moreover, this effect is stronger when $k$ is relatively small compared to $n$, which is common if we optimize for sparse masks.

**Remark** However, we will see in Section 5, that in general, the similarity of targets does not tend to be associated with higher mask overlaps if we optimize the masks for memory efficiency. Yet, memory efficiency is generally still promoted by source sharing. Furthermore, note that if the targets are highly sparse and overlap in the position of zeros, the above proof suggests that this joint sparsity could be exploited regardless of the source models. Next, we explore further options to save memory that go beyond target sparsity.

**Optimizing for memory** In practice, we can optimize the mask sizes for sparsity. In addition, we might want to maximize their overlap, as we do not have to save identical entries multiple times. What should be our optimization objective then? It should depend on the union of masks:

$$\left(\bigcup_{k=1}^{m} M^k\right)_j = \begin{cases} 1 \text{ if } \exists k \text{ s.t. } M_j^k = 1, \\ 0 \text{ else,} \end{cases}$$

where $j$ is the general index of the mask. The minimization of the union of masks takes into account the size of all masks and the overlap, making it a natural objective to minimize:

$$\min_{M^1, \ldots, M^m} \left| \bigcup_{k=1}^{m} M^k \right|, \qquad \text{s.t. } \{M^k\} \text{ satisfies Eq. (2).} \tag{4}$$

With respect to this objective, an approach that sparsifies each mask independently often leads to suboptimal results, as we discuss next.

**Theorem 3.5** *Consider the setting of Theorem 3.3. Minimizing the size of each mask $M^k$ for $k \in [m]$ is an upper bound for Eq. (4).*

Proof. It follows from the union bound that $\left| \bigcup\limits_{k=1}^{m} M^k \right| \leq \sum_{k=1}^{m} |M^k|$. This implies that

$$\min_{M^1, \dots M^m} \left| \bigcup_{k=1}^{m} M^k \right| \leq \min_{M^1, \dots M^m} \sum_{k=1}^{m} |M^k| = \sum_{k=1}^{m} \min_{M^k} |M^k|$$

where $M^k$ for all $k \in [m]$ satisfies the constraint in Eq. (2). $\square$

Theorem 3.5 implies that optimizing each mask individually does not necessarily lead to memory savings. We can thus improve upon this by considering other objectives such as the union of masks directly. The next section explores this reasoning in the context of subsetsum approximations.

## 4 INSIGHTS INTO SUBSET-SUM APPROXIMATION

Let us consider a variant of the subsetsum approximation problem with $m$ target models of length $p$. As we saw in Theorem 3.1, we can approximate each target of a model with the same source set. We will now have $p$ source sets of size $n$ and $m$ target models. This gives us a target matrix $Z \in [-\frac{1}{2}, \frac{1}{2}]^{p \times m}$, mask $M \in \{0, 1\}^{p \times m \times n}$ and source networks $X \in U[-1, 1]^{p \times n}$. Then we want to optimize the subsetsum problem for each target, given a source network. We refer to this problem as the **multi-subsetsum approximation** problem.

**Improving memory savings**    In the previous section, we have seen how that we only need a single source set to represent $m$ different targets with no additional width requirements. Since the widely used random number generators are deterministic with respect to the seed $s$, we only need $s$ and the binary subset masks to reconstruct target values to a certain precision $\epsilon$. Since binary matrices require lesser memory compared to floating point parameters of the model, this is an efficient alternative to compress multiple models of the same architecture during deployment. In addition, efficient algorithms like bitwise compression, run-length encoding, etc. can be used to efficiently store sparse binary matrices. Having sparse matrices can improve the memory savings and also lower the inference time computations.

To save memory, we prefer smaller source set sizes $n$, sparser masks $M$, and also care about a low computational overhead to reconstruct a model. The choice of the source set size $n$ depends on the permissible target error $\epsilon$ and the probabilistic error of finding a satisfying subset $\delta$. A s $\epsilon, \delta$ decrease, the necessary source set size that satisfies the errors increases. Together, the errors have an impact on how identical the performance of the approximated network is to the target network. At the same time, for a given $X_i$ for $i \in [p]$, several subsets $M_{i,j}$ for $j \in [m]$ satisfy the target requirement. As $n$ increases, so do the choices of $M_{i,j}$. Based on how we choose subsets from the choices we can gain further memory improvements. We will next discuss different storage formulations and then evaluate approaches that pick optimal subsets according to different metrics and as a result, increase the sparsity of $M$ and reduce the number of operations to reconstruct the model.

$m$ **and** $m + 1$ **scheme:**    We discuss two schemes to store the binary masks (Figure 3). The first is the general $m$ scheme that represents the binary mask as $M$ with $m$ binary planes. Instead, the $m + 1$ scheme decomposes the masks into one overlap plane and $m$ extra planes. In the overlap plane, the bits common to all $m$ planes are set. And $m$ extra planes contain the remaining set bits. If $\hat{M}$ is the mask of $(m + 1)$ scheme, the relation between $M$ and $\hat{M}$ is:

$$\hat{M}^{m+1} = \Pi_{j=1}^{m} [M^j] \tag{5}$$

$$\hat{M}^j = M^j - \hat{M}^{m+1}, \qquad \forall j \in [m], \tag{6}$$

where $\Pi$ is the element-wise product or the Hadamard product. Similarly, we can retrieve the overlap and extra planes by using bitwise operators. Once we transform each $n$ bit binary vector in $M$ into an integer, simply replacing $\Pi$ with *bitwise and operator*($\wedge$) in Eq (5) and subtraction with *bitwise xor operator*($\oplus$) in Eq (5). We can then convert the integers into $n - bit$ binary representations ($binary(\cdot)$). Similarly, the mask $M$ can be derived from $\hat{M}$ by adding it back during inference as $M^j = \hat{M}^{m+1} + \hat{M}^j$.

A naive way of saving sparse binary matrices is to store the positions of ones (also called bits) in the mask. Since, the overlap bits are repeated by all the $m$ masks, an efficient way to improve

memory savings is to save the overlap location separately, but only ones. Computing the overlap operations also only once at inference, could also lead to computational speedups. Therefore, the $(m + 1)$ scheme is a better alternative to $(m)$ scheme for memory efficiency, provided that the approximating subsets have a sizeable number of overlap bits. Also if the entire model needs to be loaded into memory (at inference), $(m + 1)$ is preferable. But in situations where the degree of overlap is not high or when only one individual model out of $m$ needs to be loaded (like on edge devices with low memory), the advantages of $(m + 1)$ vanishes as the presence of the overlap layer is redundant. Which scheme should be preferred thus depends on the most common use cases.

Since mask structure impacts memory and inference efficiency, the natural followup question is to ask whether it can be optimized. This is possible when multi-subset-sum approximation results in multiple choices for each of the $m$ targets. While reducing the joint memory footprint, we still want the sparsity of the individual masks to remain high for fast inference. With these requirements in mind, we propose the following subset selection approaches.

**Sparsity by fixed binning** ($Partition$)    In this first approach, we split the range of the target values $[-\frac{1}{2}, \frac{1}{2}]$ into fixed individual bins of size $\epsilon$ and make the center of each bin the representative target. The idea is to then map any arbitrary target value to its closest bin and pick the sparsest subset that approximates its representative. We abuse the notation and call this approximating the bin. For a given $\epsilon$ and target $z$, the representative target $z_{bin}$ can be calculated as:

$$z_{bin} = \left( \lfloor \frac{z + \frac{1}{2}}{\epsilon} \rfloor + \frac{1}{2} \right) \cdot \epsilon - \frac{1}{2} \tag{7}$$

Alg. 1 reports how we solve the associated **multi-subset-sum** approximation problem and Alg. 2 states the binning approach. Picking the sparsest subset for each target encourages sparsity across $m$ planes. It is important to note that the subset for each target is still picked independently of other targets. So, the overlap is not taken into account. This approach is catered towards the $m$ scheme.

**Sparsity by local binning** ($Local - bin$)    Even though the prior strategy to create fixed bins provides a practical framework to the problem of picking ideal subsets, it is not necessary and the mapping is an avoidable overhead, as we will see in Sec (5). For all practical purposes, it could be sufficient to assume a local bin of width $2 * \epsilon$ centered around the target value. Similar to the prior approach, sparsity can be independently maximized among approximating subsets. The algorithm is the same as Alg. 2 but without the target mapping in Alg. 1, thus also catering to the $m$ scheme.

**Minimal total length** ($Local - bin + opt$)    The earlier two approaches only promote high mask sparsity but do not focus on giving structure to $M$. This approach addresses both considerations by minimizing the total length: overlap bits $+ \sum_m$ extra bits. This promotes the selection of subsets with a higher overlap compared to the earlier approach, which did not take this into account. Note that the ultimate aim is to minimize total length, which also restrains set overlap bit sizes.

We use local binning strategy for this approach. The algorithm for optimal total length is shown in Alg. 3. Firstly, We find all possible unique overlap patterns ($overlaps$) across subsets that satisfy each of $m$ targets. Then, for each overlap ($o \in overlaps$), we find the candidate subsets ($ind_i$), the ones that possess the overlap pattern, and pick the ones with the smallest number of extra bits ($setBits(o \oplus ind_i)$). In terms of bit-wise operations, $ind_i$ has the $o$ pattern if $(o \wedge ind_i) = o$. We then pick the $(o, \{ind_i, \forall i \in \{1, 2, ..., m\}\})$. that has the minimum *total length*$= [setBits(o) + \sum_{i=1}^{m} setBits(o \oplus ind_i)]$. Here the function $setBits(\cdot)$ counts the number of bits that are set in the equivalent binary representation. Finally, we choose the combination of (overlap,candidate subsets with minimal length) which has the lowest total size.

## 5 EXPERIMENTS

To verify the extent to which memory and compute savings are possible, we run 10,000 iterations of **Multi Subset-sum** approximation according to Alg. 1] across $m$ different targets where $Z$ and $X$ are sampled uniformly at random from their respective ranges as in Theorem 3.1. We compare the total bits required for both the $(m)$ and $(m + 1)$ plane scenarios, the overlap bits and extra bits across different values of $\epsilon$, source sizes $n$ and in the cases of shared and different source sets for

---

**Algorithm 1** Multi-subset-sum

---

**Require:** Source set $X \subset (-1, 1)$, $m$ target values $Z \subset (-\frac{1}{2}, \frac{1}{2})$, allowed error $\epsilon$, optimizing algorithm $OptAlgorithm$

  **if** $OptAlgorithm$ = Partition **then**

    $Z \leftarrow [floor(\frac{Z + \frac{1}{2}}{\epsilon} + \frac{1}{2})]\epsilon + \frac{1}{2}$

  **end if**

  Initialize each element of $subsetSums$ of size $2^n$ to 0

  **for** $i \in \{1, 2, ..., 2^n\}$ **do**

    $subsetSums[i] \leftarrow X \odot binary(i)$

  **end for**

  **for** $i \in \{1, 2, ..., m\}$ **do**

    $satIndices[i] \leftarrow \{s \mid s \in \{1, 2, ..., 2^n\} \ \& \ |subsetSums[s] - Z[i]| < \epsilon\}$

  **end for**

  $optSubsets = OptAlgorithm(satIndices)$

  **return** $optSubsets$

---

**Algorithm 2** Local-bin

---

**Require:** $satIndices$ for $Z$, the number of targets $m$

  **for** $i \in \{1, 2, ..., m\}$ **do**

    $ind_i \leftarrow \arg\min_{j \in \{1,2,...,|satIndices[i]|\}} setBits(satIndices[i])$

  **end for**

  **return** Optimal subsets $\{binary(ind_1), binary(ind_2), ..., binary(ind_M)\}$

---

targets. In addition, we also compare the average number of computations required to reconstruct the $Z$. We can use the number of set bits of in $M$ and $\hat{M}$ as a surrogate for computations of the $(m)$ and $(m + 1)$ plane scenarios respectively. We refer to the approach of finding sparsest subsets by mapping targets to bin representatives as $Partition$, the approach of finding sparse in a localized bin around the targets directly as $Local - bin$, and the approach of optimizing the overlap and extra bits in the $(m + 1)$ plane scenario as $Local - bin + opt$ in the plots.

The statistics for $n = 17$ (where $\delta \sim 0.01, \epsilon = 0.001, m = 4$) for shared and different sources are shown in Figure (1). It is important to note that both *Partition & Local-bin* behave similarly across different metrics. *Local-bin* is the better option, since it removes the need to map targets to bin centers according to Eq (7).

$(\mathbf{m})$ **vs** $(\mathbf{m + 1})$ **plane scenario:** As intended, approaches like *Partition & Local-bin* have better total bits and average computations than *Local-bin+opt* in the $m$ plane scenario. Since the number of overlap bits for the subsets found by the former approach are close to zero, having an additional extra plane only adds redundancy. Whereas, *Local-bin+opt* seems to find subsets where the number of set overlap bits is much higher and is comparable to the average extra bits per target($\frac{\text{extra bits}}{m} \approx 2.5$). In addition, the number of extra bits is also lower. Since decomposing binary mask into overlap and extra bit planes in such cases improves the overall sparsity in the extra planes when the model is saved according to the $(m + 1)$ plane scheme, we see that the number of total bits and average computations required to reconstruct network to a suitable approximation are lower. This makes *Local-bin+opt* with masks saved according to $(m+1)$ plane scenario, the best methodology to avail for ensemble models or if an entire MoE model must be loaded onto memory at once. On the other hand, when models should be deployed on edge devices with low memory, loading and evaluating only the required expert head of an MoE model is more prudent. This can be quantified by the average computations in the $(m)$ plane scheme and hence *Local-bin* is the recommended approach here.

**Shared source vs different sources:** As shown in Fig 1, having a shared source increases overlap bits and reduces extra bits compared to different sources, confirming the intuition acquired from Theorem 3.4. The effect is more pronounced for *Local-bin+opt* where having a shared source lowers both the total bits required and average computations over the $(m + 1)$ plane scenario to which the approach is better suited. On the other hand, there is no tangible difference between using a shared source or different sources to approximate targets for *Partition & Local-bin*. This is as expected

---

**Algorithm 3** Local-bin+opt

---

**Require:** $satIndices$ for $Z$, the number of targets $m$

$overlaps \leftarrow \wedge_{i=1}^{m} s_i, \forall$ combinations of $s_i \in satIndices[i]$

$optOverlap, optInd \leftarrow \arg\min_{o \in overlaps, ind_i \forall i \in \{1,2,...,m\}}[setBits(o) + \sum_{i=1}^{m} setBits(o \oplus ind_i)]$

such that $ind_i \in satIndices[i]$ & $[(o \wedge ind_i) = o]$, $\forall i \in \{1, 2, ...m\}$

**return** Optimal subsets $\{binary(optInd_i), \forall i \in \{1, 2, ..., m\}\}$

---

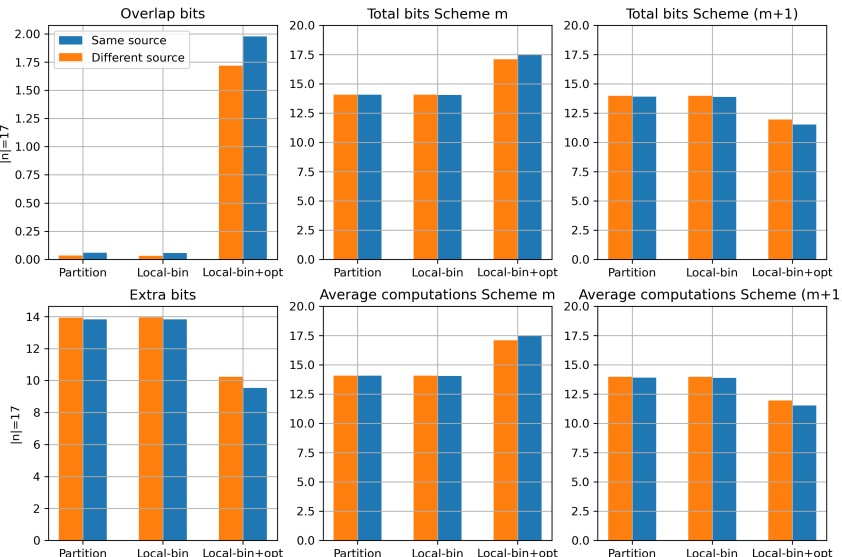

Figure 1: Memory savings and inference speedup for the proposed sparsity approaches over shared and different sources. The first column shows the average number of overlap and extra bits required by each approach, the second and third columns show the total number of bits to be saved and the number of computations to be performed at inference following the $m$ and $m + 1$ schemes.

as each target is optimized on its own. In addition, we show in the appendix that for different $n$, the success probability $(1 - \delta)$ is higher for shared source than different sources. Therefore, it is preferable to use a shared source as a general rule.

**Impact of source size $n$:** The statistics for $n = \{15, 16, 17\}, \epsilon = 0.001, m = 4$ where source set is shared are shown in plot Fig. 2. As the source size increases, the overlap among chosen subsets increases and the total bits to approximate targets and the computations decrease irrespective of the $(m)/(m + 1)$ scheme. Also having a higher $n$ reduces the error probability $\delta$ since $n \geq C \log \frac{2}{\min\{\epsilon, \delta\}}$ from Theorem 3.1. Yet, using a higher $n$ comes at a cost. Solving the **multi-subset-sum** approximation (as shown in Alg. 1]) problem requires searching for ideal subsets in a search space of size $2^n$. Therefore, it can be computationally expensive to find binary masks.

**Reliable reconstruction using SLTs:** Next, we verify if the subsets found by the proposed approaches can approximate a target MoE following the explicit SLT reconstruction approach (Burkholz, 2022a). As the target network, we consider an MoE model with 8 experts, each a single-layered 1-dimensional convolutional network, and a single-layered convolutional layer as router network. The MoE model is trained on the dataset created according to Definition 3.1 of Chen et al. (2022) and has an accuracy $\sim 94.1\%$. We solve the multi-subsetsum problem over the randomly initialized parameters of a source network with 2 layers by comparing our different approaches. We use $n = 15, \epsilon = 0.01$, which guarantees $\delta < 0.01$. Note that, for expert models with $L = 1$, the reconstruction is the same with both the 2L (Pensia et al., 2020) or L+1- constructions (Burkholz,

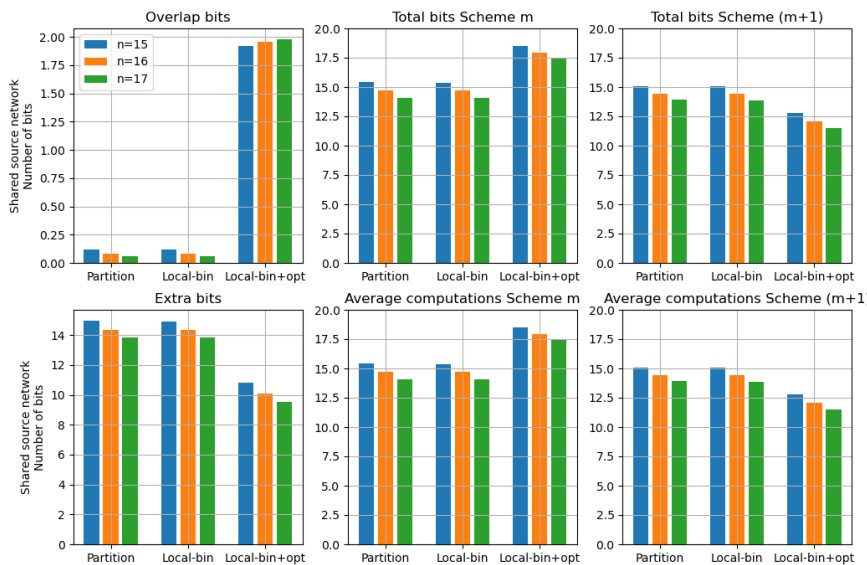

Figure 2: Memory savings and inference speedup over different sizes ($n$) of shared source network. The first column shows the average number of overlap and extra bits required by each approach, the second and third columns show the total number of bits to be saved and the number of computations to be performed at inference following the $m$ and $m + 1$ schemes.

2022a). We find that every approach is able to find masks that ensure robust reconstructions of the target network with accuracy $\sim 94\%$ over both shared and different source networks.

# 6 DISCUSSION

We investigate the potential of the strong lottery ticket (SLT) concept to reduce the joint memory footprint of model families that have similar deep neural network architectures and have SLT approximations, which can be subnetworks of the same source network. If the models overlap in their sparsity patterns in addition, they can be further exploited for compressed representations, if they share the same source network in the SLT construction, as we have established theoretically. Beyond similarity in the original parameterization, we have propose and analyze multiple criteria to select jointly optimized masks that characterize individual models, which we derive from novel insights into subset-sum approximation. This enabled us to identify fundamental trade-offs between computationally relevant objectives like the inference cost and memory footprint. At first consideration surprisingly, we further find that larger source sizes, which imply larger masks, can achieve higher memory savings, as they offer more masks to choose from during the optimization. Beyond a certain source size threshold, however, we expect that this effect saturates so that increasing the overparameterization further can only have a negative impact. Yet, this threshold is not attainable in a computationally effective way because the space of all potential subsetsum approximations increases exponentially in the source set size. However, we identify potential for already great memory improvements for moderate, practical sizes of source sizes. With our analysis, we have thus laid the theoretical basis for utilizing the SLT principle to obtain memory savings of multiple large-scale models.

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

## A  SCHEMATIC DIAGRAM FOR THE TWO MASK SAVING METHODS

We provide a schematic diagram shown in 3 to explain the $(m)$ and $(m+1)$ methods to save subsets masks.

## B  OTHER MEMORY SAVING ALGORITHMS

In this section we present additional experiments on memory savings. We will consider the case where we have two target models and look at the relationship between overlap in bits and the distance between the two targets. We consider also multiple different algorithms. The algorithms we consider are:

- Smallest superset: Minimizes the superset from which all masks have too choose. This promotes overlap by shrinking the total posibilities for each mask.
- Smallest overlap + extra (Local-Bin + opt): See Section 4.
- Smallest extra - overlap: Minimizes the extra bits used and maximizes the shared bits.
- Smallest extra: Minimizes the extra bits used.
- Local-Bin: See Section 4.
- Partition: See Section 4.

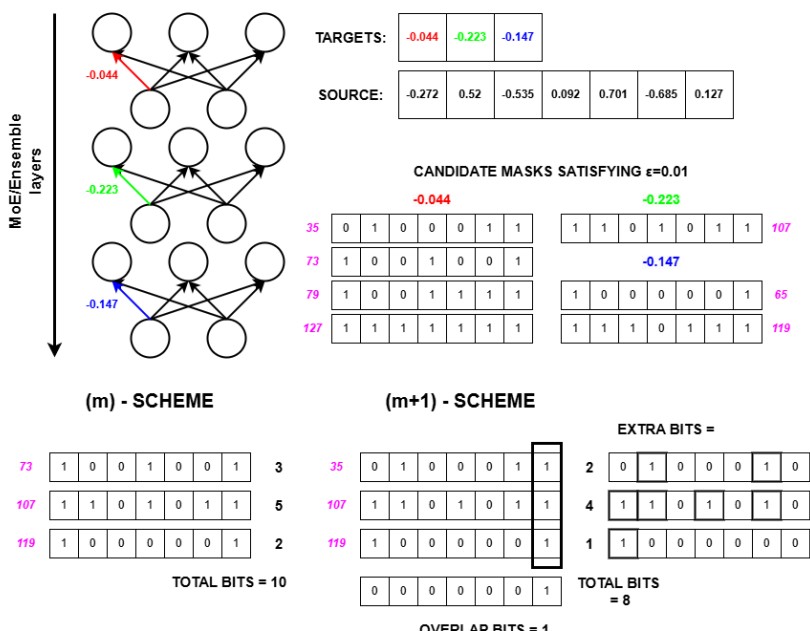

Figure 3: Schematic diagram for the $(m)$ and $(m+1)$ mask saving methods. The numbers in pink are the integer equivalents of the binary subset(e.g., 0100011 = 35). While the $m$-scheme individually picks the sparsest candidates to represent each target, $(m + 1)$-scheme picks the candidates that minimize the sum of the set overlap bits and extra bits.

Note that all algorithms take the best representative for each mask in a $(-\epsilon, \epsilon)$ interval for their respective optimization task. In contrast, partition, splits up the target space $[-\frac{1}{2}, \frac{1}{2}]$ into sub intervals of size $2\epsilon$. Then for each target it takes the best representative that corresponds to the subinterval. Moreover, we use single source size $n = 14, 15$ and $16$ .

The results for the distance versus overlap is presented in Figure 4. We observe that for all considered algorithms the distribution is uniform. Furthermore, note that between methods there is still a significant difference as also observed between the algorithms considered in the main text.

The results for the storage of overlap, extra and total bit storage is given in Figure 5. Note that we here are in the same setting as Figure 2 and use $n = 15, 16$, and $17$. We observe that the method that maximize overlap and the method that minimizes extra bits have the largest bit overlap, followed by local-bin+opt. Nevertheless, maximizing overlap or minimizing extra bits comes at a cost in the $m$ scheme, while not punished in the $m + 1$ scheme. Furthermore, as expected, local-bin+opt still uses less bits in $m + 1$ scheme. Moreover, individual optimizers such as partition and local-bin are best in the $m$ scheme and competetive with the extra proposed methods in the $m + 1$ scheme.

## C  EMPRIRICAL ANALYSIS

In this section, we evaluate the mask savings schemes and algorithms discussed in Section 4 over their reconstruction quality and the computational and storage requirements. For the experiments, we use the largescale switch-base-8 transformer (Fedus et al. (2022a)) model that is fine-tuned for the Samsum dialogue dataset for abstractive summarization(Gliwa et al. (2019)). The model consists of 24 mixtures of experts(MoE)-layers with 8 experts in each layer. Each expert is a dense layer with $\approx 2.36$ million parameters.

For the multi-subset-sum approximations(MSSA), we choose a source set size $n = 15$, with a permissible error, $\epsilon = 0.01$. The cases where a suitable subset that satisfies $\epsilon$-approximation does not exist (can occur by a probability of $\delta = 0.01$), we choose the subset with the smallest approximation error. Since our theory applies to parameter targets in the range $(-0.5, 0.5)$, we scale the parameters

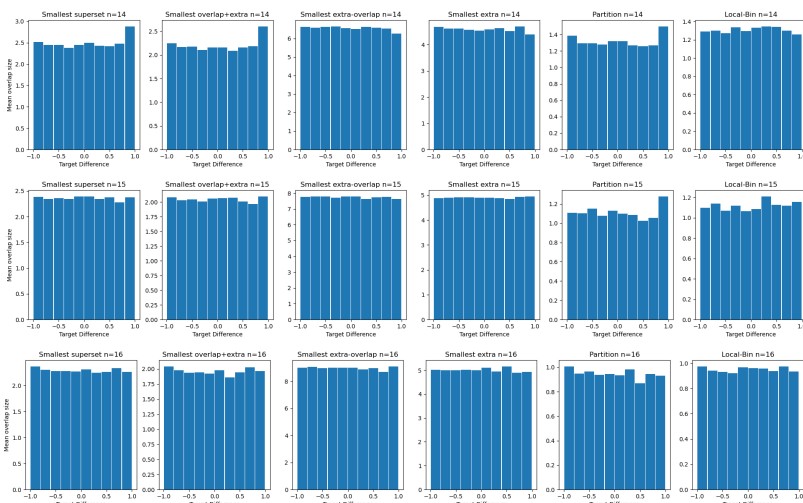

Figure 4: For all considered algorithms and $n = 14, 15$ and $16$ we show the average overlap between two targets depending on their distance to each other.

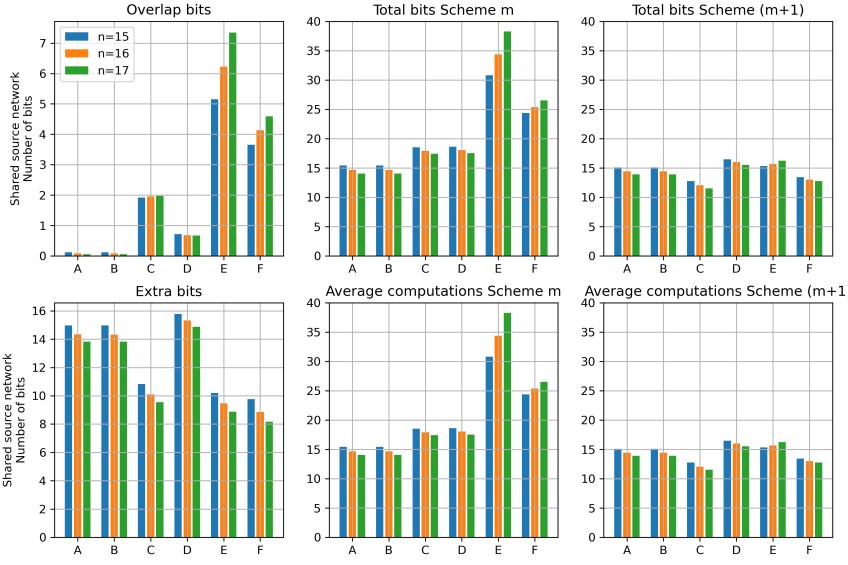

Figure 5: Memory savings and inference speedup for the additional proposed sparsity approaches over different sizes $(n)$ of shared source network. The first column shows the average number of overlap and extra bits required by each approach, the second and third columns show the total number of bits to be saved and the number of computations to be performed at inference following the $m$ and $m + 1$ schemes respectively. A-Partition, B-Local-bin, C-Local-bin+opt, D-opt(overall subset), E-opt(extra-overlap), F-opt(extra)

of each expert layer to this range. Doing so would not be inappropriate since the model architecture uses a ReLu activation function which is scale-invariant in its behavior. Also, since most activation functions can be approximated using ReLUs (Burkholz (2022a)), this scaling can be applied in general. We try to approximate the 8 target values using source sets sampled uniformly from $(-1, 1)$. All computations to find the approximating subsets are performed on a single 2nd-generation Intel Xeon Scalable processor (3.9GHz).

## C.1 RECONSTRUCTION QUALITY

We compare the text summarization performance of models reconstructed using our proposed algorithms over the Rouge suite metrics. The detailed results are presented in Table 1. Compared to the original model, the models reconstructed using both the $Partition$ and $Local-bin$ show negligible degradation to the summarization performance over the validation set of the Samsum dataset. This confirms the qualitative viability of the proposed methods without the need for excessive SLT sizes. Since they are well-optimized for the task, we use the parameters $n = 15$ and $\epsilon = 0.01$ for further quantitative analysis in Section C.2.

We are unable to include the reconstruction performance of Local-bin+opt algorithm due to the high computational complexity of the Local-bin+opt approach. This limitation is addressed in detail in Section C.2.

| Metric | Original Model | Local-bin | Partition |
|---|---|---|---|
| Validation loss | 1.461 | 1.470 | 1.468 |
| Rouge-1 | 0.46539 | 0.465144 | 0.466098 |
| Rouge-2 | 0.232704 | 0.229176 | 0.227615 |
| Rouge-L | 0.394087 | 0.393255 | 0.39309 |
| Rouge-L sum | 0.430643 | 0.427835 | 0.429367 |

Table 1: Reconstruction quality of $Local-bin$, and $Partition$ algorithms compared to a full precision switch-base-8 model (Fedus et al. (2022a)). Rouge metrics are calculated on the validation set of Samsum dataset.

## C.2 COMPUTATION TIME AND STORAGE EFFICIENCY

Traditional strong lottery ticket finding process involves sampling the weights of an entire network at random and finding masks that approximate target parameter values by performing multiple subset-sum approximation operations(SSA). We mimic the same and sample *different source sets* randomly for each target parameter vector of size m=8. A multi-subset-sum approximation(MSSA) is then performed to identify ideal subsets that optimize some objective(sparsest individual subsets, total set bits length in $(m + 1)$-scheme, etc.).

**Time complexity of algorithms:** MSSA comprises two operations. Firstly, we find candidate sets that approximate each of $m$ targets by parsing through the $2^n$ possible subsets. The time complexity of repeating this for each of $p$ target vectors in an MoE layer is $O(p.2^n)$. Secondly, we choose the ideal subset. Finding the sparsest individual subsets like in the case of $Partition$ and $Local-bin$ does not incur any comparable time complexity. Whereas, $Local-bin+opt$ incurs a lot of additional overhead that scales exponentially with the number of experts $m$. Let $c$ be the average number of candidates that approximate each target to $\epsilon$. Then the additional overhead is $O(c^m)$. The encoding times per an 8 expert MoE layer observed for the algorithms and their corresponding number of set bits are shown in Table 2. To ensure a fair comparison between algorithms we consider them in the memory storage scheme that they are tailored for. Therefore, for $Partition \& Local-bin$, we follow $(m)$-scheme and for $Local-bin+opt$, we follow $(m+1)$-scheme.

As we see in Table 2, both operations are costly over millions of parameters. Especially, comparing across candidate sets of 8 targets to find subsets that optimize total length in $(m + 1)$-scheme seems to be more computationally intensive than generating candidate sets. But as a result, we do observe an improvement in the number of set bits per parameter. While we acknowledge that 4hrs. is a significant amount of time to encode, it is still negligible compared to the hundred/millions of GPU hours that large foundation models train for(Marcińczuk (2023)). Even finetuning them takes

multiple GPU hours. Also, the algorithms that we propose are post-hoc techniques that only need to be performed once.

**Multiple source sets vs single source set:** The time complexity due to the MSSA operations can be eliminated to a large extent by reusing a single source set to approximate all target parameters. Theorem 3.1, ensures the viability of this proposal since the entire target range can be approximated to a precision $\epsilon$, with a probability $\delta$. Further, the $Partition$ algorithm serves as a perfect match since we need only calculate candidates for each of the $1/epsilon$ bins once and choose the corresponding sparsest representation. Essentially, this creates a lookup table that can be reused across all target parameter vectors thereby, accelerating the encoding process of the entire model by leaps and bounds to mere seconds. But, a tradeoff exists between the encoding time and the number of set-bits within subset representation per parameter as shown in Table 2.

|  | Multiple source sets | | | Single source set |
|---|---|---|---|---|
|  | Partition | Local-bin | Local-bin+opt | Partition |
| Encoding time/layer | 4 hrs. | 4 hrs. | 23 days* | 0.424 sec. |
| Set-bits/parameter | 1.7438 | 1.7436 | 1.5740 | 2.4 |

Table 2: Comparison of encoding times and set-bits per parameter when multiple source sets vs a single source set are sampled to approximate parameters. For a fair comparison, the set-bits/parameter is calculated according to $m$-scheme for both $Partition$ and $Local - bin$ and $(m + 1)$-scheme for $Local - bin + opt$ algorithms respectively.

**Storage Compression Formats:** To analyze the extent to which the mask sparsity and similarity can be exploited, we compare four different compression/file storage schemes. They are:

- **Float-32:** This is the default way models are saved where each parameter is stored as 32-bit float data. This will serve as the baseline.

- **N-bit Mask:** The n-bit masks, that are identified by the multi-subset-sum approximation over a source set of size $n$, and the associated seed are saved.

- **Huffman Encoded - Set Bit Positions(HE-SBP):** The main goal of this storage format is to save the positions of set bits within the n-bit mask efficiently and to exploit the sparsity of the binary masks. We use Huffman encoding to optimize the binary code representation of each of the $n$ bit positions based on the frequency with which they occur across all parameters in an MoE layer. It is important to note that a delimiter token must also be included within this style of encoding since the number of set bits within approximating subsets of different parameters can vary. Within the encoded file, a delimiter marks the end of a list of set bit positions applicable to the parameter that is being parsed. HE-SBP can be applied to store masks following either the $(m)$-scheme or the $(m + 1)$-scheme and any of the three proposed algorithms. It is important to note that the code block that compresses a target vector of size $m$ would have $m$ delimiter tokens in the $(m) - scheme$, whereas, $(m + 1)$ delimiter tokens in the $(m + 1)$-scheme. To decode, one must save the seed, the encoded message as well as the Huffman tree that was used.

- **Huffman Encoded - Bin Representatives(HE-BR):** Using Huffman encoding, this method encodes the unique approximating subset masks directly. It couples well with the masks found by the Partition algorithm operating on a single source set for all target parameter vectors. This is because the number of unique approximating subset masks are restricted (= $1/epsilon$ bin representatives). Its compression performance suffers as the number of unique approximating subsets increases.

The storage requirements and the time to encode and decode based on the methods discussed above are shown in Table 3. We have tried a combination of mask-finding algorithms(in their optimal mask-saving schemes), single vs multiple sources, and shared vs different sources that were relevant to the storage formats.

As expected, the *15-bit mask* was able to reduce the storage requirement by half with no significant computational overhead to encode/decode. However, a loose trade-off exists between the storage

| Method | Storage Space | Bits/ Parameter | Encode Time (Full model) | Decode Time (Full model) |
|---|---|---|---|---|
| **Float-32** | 1.728 GB | 32 | - | - |
| **15-bit mask** | 810.01 MB | 15 | 10.18 sec | 16.32 sec |
| **HE–SBP** | | | | |
|   Partition Single source | 602.74 MB | 11.16 | 589 sec | 4065 sec |
|   Partition Multiple sources | 527.1 MB | 9.46 | 4 days | 4150 sec |
|   Local-bin+opt Multiple sources | 556.9 MB | 9.99 | 23 days* | 4133 sec |
| **HE–BR** | | | | |
|   Partition Single-Shared source | **265.82 MB** | **4.92** | 331 sec | 828 sec |
|   Partition Single-Different sources | 292.75 MB | 5.31 | 399 sec | 1102 sec |

Table 3: Comparison of various storage compression formats. Note: Partition follows $(m)$-scheme and Local-bin+opt follows $(m+1)$-scheme.

requirements and computational overhead. While HE-SBP and HE-BR formats compress the model even further, they pay in terms of the compute required to encode and decode.

**Shared vs Different sources:** Overall, the combination of HE-BR + Partition on a single shared source strikes a good balance between memory and overhead. Also, *having a shared source is beneficial over having different source networks* within the HE-BR format. This reinforces our insights from Section 5. Having different sources increases the number of unique approximating subsets resulting in increased sizes of the individual Huffman codes.

**Single vs Multiple sources:** Having individual sources for each target parameter vector results in a better storage and set-bits/parameter rate than a single source. Having a single source network freezes the candidates that can approximate each bin centre. If certain bins $(m)$/combinations of bins$(m+1)$ end up having candidates that aren't sparse, it affects the overall sparsity. Whereas, having multiple source networks makes the system more dynamic and increases the chances of finding sparser candidates that can be chosen as representatives.

$(m)$ **vs** $(m+1)$**:** Counter to our findings in Section 5, optimizing mask to be sparse in an $(m+1)$ mask saving scheme doesn't seem to translate into storage benefits. Though the number of set-bits/parameter (2) of $(m+1)$-scheme is lesser than the $(m)$-scheme, the presence of additional delimiter token due to the overlap bits seems to nullify the effect. This is even though HE-SBP optimizes the code over set bit positions.

