# OpenReview forum: "Memory Savings by Sharing One Source: Insights into Subsetsum Approximation"
_ICLR.cc/2025/Conference — ICLR 2025 Conference Withdrawn Submission_

### Official Review · Reviewer_MJyp · 2024-11-01

**Soundness:** 3
**Presentation:** 2
**Contribution:** 2
**Rating:** 5
**Confidence:** 3

**Summary:**

This paper investigates Strong Lottery Tickets (SLT) to reduce memory consumption of large neural networks. Multiple models can use the same source without additional memory by using SLT, which is defined by a shared source network and uses binary masks to select specific subnetworks. The authors propose techniques to improve SLT-based memory efficiency by sharing sparsity patterns and improving subset-sum approximation (which limits memory overhead by allowing similar target models to share masks and memory resources). This approach can support ensembles and mixture of experts (MoE) architectures in memory-constrained environments.

**Strengths:**

1. SLT is an innovative approach to reduce the memory overhead of sharing source networks across model families (e.g., ensembles).

2. The proposed method may help practical applications by making large models more practical on resource-constrained devices such as edge devices.

3. The experiments are well designed to demonstrate the advantages of the proposed techniques over the current techniques.

**Weaknesses:**

1. Determining the optimal binary mask and subset sum may result in higher computational overhead, which may outweigh the memory savings.
2. Although the research touches on certain practical issues (such as redundancy in sparse models), more examination of failure scenarios and edge cases could improve the work.

**Questions:**

1. How does the approach perform on heterogeneous models that lack similar sparsity patterns?
2. What are the scalability implications of the subset-sum approximation approach for larger datasets or more complex models?
3. Can the proposed binary mask optimization process be streamlined to reduce computational overhead, especially in dynamic environments?

---

> ### Author Response · Authors · 2024-12-03
>
> Thank you very much for your valuable reviews. As we realized, the requested large scale experiments take longer time than we had during the rebuttal period. For this reason, we have decided to retract the paper. However, we will integrate your feedback into a resubmission.

---

### Official Review · Reviewer_AbBP · 2024-11-03

**Soundness:** 3
**Presentation:** 3
**Contribution:** 4
**Rating:** 8
**Confidence:** 3

**Summary:**

This paper proposes a novel strategy to reduce memory usage of multiple target neural networks with the same or similar architectures, for example, specialized models fine-tuned from the same foundation model or Mixture of Experts (MoEs) popular in Large Language Models (LLMs). To achieve the above goal, the authors apply and extend the strong lottery ticket (SLT) concept to storage of multiple networks by sharing a common source network and using target network specific mask. Theoretical analysis and experimental results show that the proposed method can reduce memory usage and speed up inference. Considering future deployment of multiple specilized and operative AI models for complex tasks, this work offers an interesting and promising solution for effective model storing and running.

**Strengths:**

The idea of applying the SLT concept to model storage is creative and interesting. Sharing a common source network and encoding each individual model as a binary mask enables a compact storage of multiple models and even speeds up inference by reusing overlapping computations. The authors provide construction algorithms for identifying the masks.

The work is valuable for building systems driven by multiple specialized AI models.

The paper is well organized but not so easily understood.

**Weaknesses:**

The text should be more readable if some prelimiary knowledge is given.

**Questions:**

1. A schematic digram should be very useful for readers to grasp the whole picture and key details of this work.

---

> ### Author Response · Authors · 2024-12-03
>
> Thank you very much for your valuable reviews. As we realized, the requested large scale experiments take longer time than we had during the rebuttal period. For this reason, we have decided to retract the paper. However, we will integrate your feedback into a resubmission.

---

### Official Review · Reviewer_iLTV · 2024-11-04

**Soundness:** 1
**Presentation:** 1
**Contribution:** 1
**Rating:** 3
**Confidence:** 5

**Summary:**

This paper proposes to use Strong lottery tickets (SLTs) to reduce memory usage of large deep neural network execution. Specifically, the author of the paper propose to use a seed for generating a random source network and a binary mask, and multiple models can share the same source network without increasing its width requirement by sharing specific sparsity patterns.

**Strengths:**

1. This work utilizes the SLT concept to reduce the memory footprint of neural network model families, which is a promising topic.

2. This work gives theoretical proof of using subsetsum approximations on the design, which is a solid approach. The experimental results also consolidate the theory.

**Weaknesses:**

1. This work is based on a problematic concept of SLT, which has many potential problem itself (e.g., long searching time, fail to improve accuracy if trained on the subnetworks, etc.). Therefore, the design and solution is not convincing.

2. This paper claims the memory reduction by using SLT. However, the experiments are not quite sufficiently showing such claim. It is not clear how memory is reduced. Also SLT can reduce memory is already proved in Otsuka et al., 2024, therefore making the contribution of this paper limited.

3. The experiments are not thorough. The author of the paper only conduct very limited experiments, with only two figures. Besides, the figures are also showing limited information. There are no quantitative data or analysis.

4. The paper has some minor writing issues. The writing is not consistent and the elaborations are sometimes confusing. For example, in line 738-739 in appendix, the reference of the section is missing with question marks.

**Questions:**

Please refer to weaknesses

---

> ### Author Response · Authors · 2024-12-03
>
> Thank you very much for your valuable reviews. As we realized, the requested large scale experiments take longer time than we had during the rebuttal period. For this reason, we have decided to retract the paper. However, we will integrate your feedback into a resubmission.

---

### Official Review · Reviewer_rQSP · 2024-11-04

**Soundness:** 1
**Presentation:** 2
**Contribution:** 3
**Rating:** 3
**Confidence:** 2

**Summary:**

The authors propose to address the high memory requirements of foundation models, specifically ensembles or Mixture of Experts (MoEs), where multiple models may be derived from a single base model, by exploiting Strong Lottery Tickets (SLTs). SLTs are models with random weights for which a subnetwork (i.e. mask) identifies a strong solution to a problem. The authors propose that SLTs could provide a base for MoEs/ensembles of foundation models where the models would share a common base, but with different sparse masks. The authors propose that this could lead to memory savings, and claim to validate their theoretical findings.

**Strengths:**

* The theoretical argument that MoE, ensemble or weight-averaged models might share the same strong lottery ticket basis is interesting theoretically, and I believe a novel research direction.
* Strong lottery tickets do not technically require training, but do require very large model widths, and identifying the masks is non-trivial. The paper focuses on understanding how the requirements for allowing a shared "base model" in an MoE scenario would affect the required model width of a SLT, with an interesting claim that the width of an SLT does not need to increase over a single model.

**Weaknesses:**

* While I understand at a high level the theoretical motivations of the paper, and that strong lottery tickets themselves do not require training per-se, finding the masks for SLTs amounts to optimizing/training the masks which could be expensive at high model widths for which SLTs work. It's not obvious this method would be practical in practice even for inference at SLT widths, and it's most importantly not clear to be that the memory savings would be worth the extra computational costs. There are no discussions on even theoretical computational costs and if these could ever not be the bottleneck in an MoE or ensemble rather than memory.
* No empirical evidence, which I believe is necessary given the non-theoretical motivations and claims of the authors that the theoretical results are validated in experiments.
* Only memory savings results are the theoretical results in Figure 2 that appear to assume a straightforward reduction in memory proportional to the number of shared models.
* Multi subset sum approximations can also be expensive, and in the author's experiments, they do not discuss the computational requirements of this part of their methodology. In Algorithm 1 it appears the approximation is O(2^n).
* I found the motivation and claims on contributions towards making MoEs/ensembles more memory efficient unconvincing given the author's theoretical findings and lack of experimental validation. Although I understand these are timely and appealing motivations in themselves, the author's proposed theoretical claims have little to do with memory efficiency of ensembles and MoEs in practice. These are very dependent on hardware and implementation, especially "memory footprint", which is much more about intermediate representations than just the number of weights of the models in practice.
* The authors appear to conflate "memory footprint" and the number of weights in a model (i.e. size of a persisted model). While obviously the number of model weights do play a factor in the memory footprint of a model, it's usually a minor one compared to other factors more specific to the architecture and implementation of that architecture. It's fairly difficult to motivate methods that reduce the number of weights of a model alone as memory footprint and computational complexity are significant bottlenecks, while storage is not typically a bottleneck in contemporary compute settings/applications.
* There is a wide body of literature in deep learning related to the idea of having a base model with different sparse masks might be used to learn different representation has been proposed and explored in many settings already in numerous papers unreferenced, notably "Piggyback: Adapting a Single Network to Multiple Tasks by Learning to Mask Weights", Mallya et al., ECCV 2017, and "Many Task Learning With Task Routing", Strezoski et al. ICCV 2019. While I do not claim the author's method is not novel (I believe it is) if there are claims it would be better in practice than these previous approaches, these need to be empirically validated compared to such baselines and papers that are similar mentioned and differentiated from. If there are claims the proposed method would be theoretically better than these, perhaps that should also be demonstrated. At a minimum some of this literature should be in the related work also.
* I personally found the paper hard to understand and follow, often assuming notation or concepts without defining them or the context. For example, in the second paragraph, without great knowledge of the theoretical work in SLTs the reader would be completely lost on what a "width requirement" is or why we are interested in it here. Or on notation, simply even $\epsilon$ and $\delta$ in Theorem 3.1, 3.2 respectively, we seem to be required to assume what these are until much later in the paper.

**Questions:**

* Could the authors define clearly what they mean by "memory footprint" and how in practice this relates to memory bottlenecks for inference or training of MoEs or Ensembles on real-world hardware.
* What is the computational complexity of the methodology (i.e. both determining the SLT base/masks, **and** the inference cost of such a setup), and would the compute bottleneck of the proposed method be of concern before the memory bottlenecks it appears to be trying to solve?
* What is the experimental validation of the theoretical results referred to by the authors?

---

> ### Comment · Reviewer_rQSP · 2024-11-30
>
> Just to note here that with lack of author rebuttal, I'll be maintaining my rating.

---

> ### Author Response · Authors · 2024-12-03
>
> Thank you very much for your valuable reviews. As we realized, the requested large scale experiments take longer time than we had during the rebuttal period. For this reason, we have decided to retract the paper. However, we will integrate your feedback into a resubmission.

---

### Note · Authors · 2024-12-03

I have read and agree with the venue's withdrawal policy on behalf of myself and my co-authors.